# Targeting Muscle Regeneration with Small Extracellular Vesicles from Adipose Tissue-Derived Stem Cells—A Review

**DOI:** 10.3390/cells14100683

**Published:** 2025-05-09

**Authors:** Lucas Fornari Laurindo, Enzo Pereira de Lima, Adriano Cressoni Araújo, Victória Dogani Rodrigues, Jefferson Aparecido Dias, Marcos Barbosa Tavares Filho, Debora Aparecida Pires de Campos Zuccari, Lívia Fornari Laurindo, Maria Angélica Miglino, Eduardo Federighi Baisi Chagas, Claudemir Gregório Mendes, Rosa Direito, Vítor Engrácia Valenti, Sandra Maria Barbalho

**Affiliations:** 1Department of Biochemistry and Pharmacology, School of Medicine, Universidade de Marília (UNIMAR), Marília 17525-902, SP, Brazil; enzopereiradelima2020@gmail.com (E.P.d.L.); adrianocressoniaraujo@yahoo.com.br (A.C.A.); jeffersondias@unimar.br (J.A.D.); smbarbalho@gmail.com (S.M.B.); 2Laboratory for Systematic Investigations of Diseases, Department of Biochemistry and Pharmacology, School of Medicine, Universidade de Marília (UNIMAR), Marília 17525-902, SP, Brazil; 3Postgraduate Program in Structural and Functional Interactions in Rehabilitation, School of Medicine, Universidade de Marília (UNIMAR), Marília 17525-902, SP, Brazil; mbtfilho@gmail.com (M.B.T.F.); miglino@usp.br (M.A.M.); efbchagas@unimar.br (E.F.B.C.); crgremen@yahoo.com.br (C.G.M.); 4Department of Biochemistry and Pharmacology, School of Medicine, Faculdade de Medicina de Marília (FAMEMA), Marília 17519-030, SP, Brazil; vic8dr@gmail.com; 5Department of Molecular Biology, School of Medicine, Faculdade de Medicina de São José do Rio Preto (FAMERP), São José do Rio Preto 15090-000, SP, Brazil; debora.zuccari@famerp.br (D.A.P.d.C.Z.); liviaflaurindo@hotmail.com (L.F.L.); 6Postgraduate Program in Animal Health, Production and Environment, School of Veterinary Medicine, Universidade de Marília (UNIMAR), Marília 17525-902, SP, Brazil; 7Department of Animal Anatomy, School of Veterinary Medicine, Universidade de Marília (UNIMAR), Marília 17525-902, SP, Brazil; 8Laboratory of Systems Integration Pharmacology, Clinical and Regulatory Science, Research Institute for Medicines, Universidade de Lisboa (iMed.ULisboa), 1649-003 Lisbon, Portugal; rdireito@ff.ulisboa.pt; 9Postgraduate Program in Movement Sciences, Universidade Estadual Paulista (UNESP), Presidente Prudente 19060-900, SP, Brazil; vitor.valenti@unesp.br; 10Department of Biochemistry and Nutrition, School of Food and Technology of Marília (FATEC), Marília 17500-000, SP, Brazil; 11Research Coordination, UNIMAR Charity Hospital, Universidade de Marília (UNIMAR), Marília 17525-902, SP, Brazil

**Keywords:** adipose tissue-derived stem cells, small extracellular vesicles, skeletal muscles, sarcopenia, aging-related muscle loss, muscle regeneration

## Abstract

Extracellular vesicles (EVs) are membrane-bound structures released by cells carrying diverse biomolecules involved in intercellular communication. Small EVs are abundant in body fluids, playing a key role in cell signaling. Their natural occurrence and therapeutic potential, especially in the context of muscular disorders, make them a significant area of research. Sarcopenia, characterized by progressive muscle fiber loss, represents a pathological state in which EVs could offer therapeutic benefits, reducing morbidity and mortality. Recent studies have proposed an interplay between adipose tissue (AT) and skeletal muscle regarding sarcopenia pathology. AT dysregulation, as seen in obesity, contributes to skeletal muscle loss in a multifactorial way. While AT-derived stem cell (ATDSC) small EVs have been implicated in musculoskeletal homeostasis, their precise action in muscle regeneration remains incompletely understood. In this context, ATDSC-derived small EVs can stimulate skeletal muscle regeneration through improved proliferation and migration of muscle cells, enhancement of muscular perfusion, improvement of tendon and nerve regeneration, stimulation of angiogenesis, and promotion of myogenic differentiation. However, they can also increase skeletal muscle loss. Notably, this is the first comprehensive review to systematically examine the role of ATDSC-derived small EVs in sarcopenia.

## 1. Introduction

Extracellular vesicles (EVs) are cell-derived membrane-surrounded vesicles that carry various types of molecules [1,2]. These can naturally occur, but can also be engineered under laboratory settings via extrusion. EVs are released from cells and delimited by a lipid layer. EVs do not replicate independently since they do not contain any functional nucleus. The term EV must be used cautiously, and different EVs are separated based on size, density, cellular origin, or molecular composition. According to the Minimal Information for Studies of Extracellular Vesicles (MISEV) guidelines, based on their sizes, EVs are generally classified into small EVs (<200 nm in diameter) and large EVs (>200 nm in diameter) [3,4]. Small EVs drive numerous biological processes by transporting signaling molecules, including tumor metastasis, cell signaling, immune responses, and other cellular activities. Due to their ubiquity, recent investigations have proven that small EVs possess diagnostic and therapeutic functions [5,6,7]. In this context, detecting, separating, and quantifying these entities is crucial for precise and tailored medicinal interventions, especially in muscular diseases [8,9,10].

Skeletal muscles constitute around 40% of the overall body weight and 50–75% of the body’s protein stores. Their main functions involve facilitating body mobility, generating heat through thermogenesis, protecting visceral organs, and contributing to metabolic health. They also serve as protein reservoirs. Skeletal muscles are adaptable, substantial due to their homeostatic relevance, and complex, causing heterogeneous muscular ailments, including injuries, weakness, and, principally, sarcopenia, which negatively impacts quality of life [11]. Sarcopenia is a skeletal muscle disease characterized by low muscle mass, diminished muscle strength, and impaired physical performance. Sarcopenia is an aging-related disease, and its prevalence is up to 29% in community-dwelling older adults, being higher in those elderly who are hospitalized, live in residential aged care services, have multi-morbidity (multiple chronic degenerative diseases), or are frail (vulnerability due to a prominent decline in physical or functional abilities) [12].

Sarcopenia (Figure 1) is associated with increased adverse events and outcomes, such as falls, functional decline, and mortality. Pathologically, the size and number of muscle fibers (myofibers) correlate strictly with increased inflammatory cell infiltration [13], prominent mitochondrial dysfunction [14], decreased muscle satellite cells [15], and oxidative stress [16], leading to muscle protein degradation and diminished muscle repair and regenerative capacity [17,18]. Lipotoxicity and insulin resistance are also generating factors. Genetics contributes to sarcopenia. However, fatty tissue’s influence is crucial [19,20,21,22].

Recently, studies have proposed an interplay between inflammatory adipose tissue and inflamed skeletal muscle. Adipose tissue dysregulation, like obesity, is associated with skeletal muscle loss in a multifactorial way. In individuals suffering from obesity and often living with advanced age, paracrine, autocrine, and endocrine regulation (associated mainly with organokine dysfunction) lead to the inflammation of fat and muscle, which creates a vicious cycle and a condition known as inflammaging, the primary mechanism linking adipose tissue dysfunction and sarcopenia [23,24,25,26,27]. Aged and dysfunctional fat contains infiltrated pro-inflammatory and other immune cells, intensifying systemic inflammation. In this context, fat infiltrates the skeletal muscle and promotes muscle dysfunction, lipotoxicity effects, insulin resistance, and inflammation, key aspects of sarcopenia occurrence [28,29,30]. For instance, EVs from white adipose tissue progenitor cells have been described as having a potent impaired ability to mitigate inflammaging in adipose tissue macrophages. In this context, midlife patients’ adipose tissue’s EVs lacking micro ribonucleic acid (miR)-145-5p failed to suppress L-selectin in those macrophages, thereby facilitating their M1 program via the nuclear factor kappa b (NF-κB) signaling pathway. On the contrary, EVs from young adipose tissue stem cells effectively inhibited M1 macrophage polarization, promoting beneficial effects on the overall aging process [31].

Stem cells, including mesenchymal cells, are a source of small EVs. Adipose tissue stem cells are abundant and, therefore, one primary source within the body. Since small EVs have been discovered as a major paracrine factor, including during non-cell therapy development and cell-to-cell communication, isolating small EVs from adipose tissue is imperative. In this context, adipose tissue stem cells are isolated, cultured, and centrifuged to isolate small EVs, which remain beyond cellular debris and cellular components [32,33].

EVs are essential regulators of musculoskeletal health, especially those from the adipose tissue. They are released from the adipose tissue and possess positive or negative roles in skeletal muscle homeostasis [34,35]. EVs can stimulate skeletal muscle regeneration. However, they can also increase skeletal muscle loss [36,37]. Sanz-Ros et al. [38] also demonstrated that small EVs from young adipose tissue stem cells prevented frailty, decreased epigenetic age, and improved overall health span in old mice. Small EVs induced proregenerative effects in the treated old mice. Additionally, they reduced oxidation, inflammation, and senescence markers in the older mice’s muscles. Importantly, no previous published review has comprehensively and critically addressed the roles of small EVs from adipose tissue-derived stem cells (ATDSCs) in skeletal muscle regeneration or loss. Given this gap in the literature, this review seeks to explore adipose stem cell-derived small EVs and their implications for sarcopenia and aging-related muscle loss with a mechanistic view. Our review is the first to fully assess adipose tissue-derived stem cell small EVs ’ potential against muscle disease and identify their promise based on their mechanisms of action. We comprehensively address the totality of the published studies on ATDSC small EVs against muscle degeneration, reviewing their models, interventions, mechanistic insights, targeted therapies, and potential clinical applications. By utilizing a systematic approach, we evaluate the effects of small EVs on skeletal muscle cell viability and the modulation of critical molecular pathways. Our main goal is to advance the knowledge of the role of ATDSC small EVs in muscle therapy and pave the way for its potential clinical applications.

## 2. Adipose Tissue-Derived Stem Cells

Stem cells are obtained from embryonic tissue, genetically reprogrammed differentiated somatic cells, induced pluripotent stem cells, and postnatal adult stem cells [39]. Stem cells are derived from skin, blood, skeletal muscles, bone marrow, and, most importantly, fat, and the most common type are mesenchymal stem cells originating in bone marrow [40,41]. Although isolated from almost everybody’s tissue, mesenchymal stem cells are limited due to the small amount of tissue from which they can be retrieved, and only a limited number of cells can be harvested [42]. In this context, ATDSCs have become one of the most promising populations of stem cells due to the tissue’s ubiquity, which allows for the harvest of larger quantities with less donor morbidity. Since adipose tissue is abundant throughout the human body, ATDSCs can be easily obtained from areas like the abdomen, arm, and thigh subcutaneous adipose tissue and retrieved in higher numbers. ATDSCs possess multi-lineage capacity, offering insights into potential cell and tissue engineering therapies for common human diseases and conditions, like muscle loss and sarcopenia [43].

While assessing anti-aging-based cellular therapies, the quality of the donor is a significant constraint on the success of the proposed treatment. Wang et al. [44] found that young-donor adipose tissue stem cells improved anti-aging effects by reducing immune cells and inflammation in aged mice throughout secreted immune factors. These effects were not attributed to old donors [44,45]. In a previous study, Rodriguez et al. [46] demonstrated that young-donor fast-adherent adipose stem cells induced more regenerative effects on muscles than old-donor slow-adherent stem cells.

### 2.1. Overview of ATDSCs and Mechanistic Insights into Regenerative Medicine

The most common source of ATDSCs is subcutaneous adipose tissue. The term “pre-adipocyte” designates the adipose tissue-committed progenitor cell population. These are multipotent stem cells mainly regulated by peroxisome proliferator-activated receptor gamma (PPARγ) to become adipocytes [47]. To yield ATDSCs, adipose tissue is broken down by physical or enzymatic digestion with collagenases to harvest stem cells until only the stromal vascular fraction (SVF) remains, composed of all adipose tissue’s cell types except the adipocytes, which are depleted during SVF processing [48,49]. It is worth noting that cryopreservation does not substantially decrease ATDSCs’ viability. Furthermore, the cell surface antigen profile does not significantly differ among donors of different ages [50]. Reumann et al. also identified that weight and gender do not affect ATDSC quality. However, they highlighted that the donor site significantly affects the proliferation and differentiation of ATDSCs [51]. Like mesenchymal stem cells from the bone marrow, ATDSCs do not express major histocompatibility complex class II (MHC II), human leukocyte antigen-DR isotype (HLA-DR), and express only low levels of major histocompatibility complex class I (MHC I), HLA-DR, which provides relatively immune privilege due to reduced immunogenicity. However, they also appear deficient in T lymphocytes’ costimulatory molecules [52]. ATDSCs attract attention due to their vast array of soluble mediators and EV production. These alter many cells’ biology and mediate tissues’ therapeutic effects, especially in tissue protection, repair, and regeneration [53].

### 2.2. ATDSCs’ Roles in Muscle Regeneration

Skeletal muscle regeneration depends on satellite cell function. However, due to their insufficiency during volumetric muscle loss, research on myogenic stem cells has been conducted widely. It has rapidly grown on ATDSCs for their multi-directional differentiation potential and promising mechanisms of action. As a result, ATDSCs have become vital candidates for skeletal muscle regeneration. Many factors influence ATDSC differentiation, including cytokines and growth factors. Vascular endothelial growth factor (VEGF) increases muscle formation due to accelerated angiogenesis. Insulin-like growth factors 1 (IGF-1) and 2 (IGF-2) activate satellite cells, phosphoinositide 3-kinase (PI3K)/protein kinase b (Akt) signaling, block inflammation, and promote stem cell myogenic differentiation [54]. Myoblast regulatory factors can also promote ATDSC muscle production. Myoblast-derived myoblast determination protein 1 (MyoD) and myogenic factor 5 (Myf5) mainly promote ATDSCs into myoblasts, whereas myogenin and muscle regulatory factor 4 (MRF4), downstream of myoblasts, induce mature myotube formation [55,56]. Mechanistically, ATDSCs promote muscle regeneration through increased muscle repair after injury, the generation of new myofibers, the restoration of dystrophin expression, and satellite cell enhancement, restoring myogenic cell polarity and apicobasal asymmetric division and enhancing myogenic progenitor division [57,58,59]. However, combination therapies involving ATDSCs and other cell-derived factors are also paramount. Hwang et al. demonstrated that the combination of human ATDSCs and fibroblast growth factor boosts skeletal muscle regeneration after laceration, which reflects the effectiveness of a combination therapy as a promising avenue for muscle restoration [60].

## 3. A Mechanistic Approach to Small Extracellular Vesicle Isolation, Characterization, and Mediated Communication

The following lines will briefly review small EVs’ isolation and characterization. A mechanistic approach based on specific small EV cargo will also be described. Finally, small EVs will be briefly introduced in muscle research.

Small EVs have emerged as key candidates for cell communication, redefining paracrine communication and signaling. They carry and distribute various bioactive compounds among cells, including proteins, metabolic by-products, lipids, and nucleic acids. Regarding therapy, small EVs can be transferred from host to destination cells, resulting in cellular reprogramming or epigenetic information regulation depending on their content [61,62]. Small EVs can transport metabolites, deoxyribonucleic acid (DNA) fragments, coding messenger ribonucleic acids (mRNAs), miR, proteins, lipids, and noncoding long ribonucleic acids (lncRNAs) from their origin to target cells. They can regulate intercellular communication and are used in more than 400 clinical trials worldwide [63]. Small EVs must be separated from non-small EV components, including microvesicles and apoptotic bodies, in sufficient quantities, purity, and size to improve small EV-based therapies [63]. ATDSC-derived small EVs are purified using differential centrifugation. Small EV precipitation also uses ultracentrifugation (UC) and centrifugation to precipitate supernatants and enrich the process. Flow cytometry assays characterize small EVs, and mass spectrometry identifies proteins following isolation. Then, a multiplex bead-based platform enables the detection of 37 small EV surface epitomes plus two isotype controls, including many clusters of differentiation (CDs), recombinant engineered antibody control (Rea), immunoglobulin G1 (mIgG1), and HLA-DR [64,65].

Mechanistically, how small EVs exert their functions on target cells has yet to be fully understood. However, following secretion from their parental cells and enclosed by a lipid bilayer membrane, small EVs are detectable in all body fluids with varying stability, with half-lives ranging from a few minutes to several hours until captured by target cells [66]. Small EVs are believed to act on target cells by internalization through fusion with the host cell membrane. However, receptor-mediated endocytosis, phagocytosis, macropinocytosis, small EV surface-bound components cleavage, and small EV surface-associated molecules binding to target cells’ receptors are also methods of small EV action. Under physiological and pathological conditions, small EV numbers vary extensively. Impressively, cancer cells secrete up to 1000 times more small EVs than healthy, normal cells [67,68]. Small EV fusion may occur at different locations of the target cell. Plasma membrane fusion often enables the cargo molecules to concentrate in the cytosolic layer near the target cell’s surface. It is worth noting that clathrin-dependent forms of internalization also participate in small EV targeting. Following endocytic internalization, small EVs move along the endocytic pathway and co-localize with lysosome markers, including lysosomal-associated membrane protein 1 (LAMP-1), which integrates the small EV cargo into the target cell’s cytosol. Both integrated membrane and small EV cargo molecules act in small EV recycling [68,69]. Figure 2 depicts the main types of EVs and their potential contents.

## 4. Small Extracellular Vesicles from ATDSCs in Muscle Targeting

Small EVs present high stability in circulation, low immunogenicity, and minimal toxicity. Therefore, they are promising for medicinal administration, paving the way for clinical uses [70]. The following sections will explore using small EVs for muscle targeting, reviewing the possible delivery methods to target muscle cells, and the mechanisms behind their effects. Additionally, Table 1 illustrates the current research on ATDSC small EVs targeting muscle regeneration in various health conditions.

### 4.1. Recent Advances in Delivery Methods to Target Muscles with Small Extracellular Vesicles

Using small EVs in therapy presents many advantages. Compared to small EVs, liposomes, which are also composed of a phospholipid bilayer, lack target specificity. They are vulnerable to clearance by the mononuclear phagocyte cells, and their action relies on passive accumulation while delivering drugs to specific tissues. Therefore, they require sophisticated equipment for their production, elevating prices and potential immunotoxicity, decreasing instability and half-life, and imposing challenges to clinical uses. Small EVs have emerged as a promising research subject in this context since they are natural, exhibit low immunogenicity, and impose minimal toxicity, enabling direct cargo delivery in the target cells’ cytosol [79]. Moreover, due to their natural occurrence, small EVs are more easily engineered to target specific cells through magnetic materials, pH-responsive motifs, and ligands, enhancing therapy specificity and efficacy, especially against whole organs [80]. Regarding delivery, small EVs can be modified to deliver to specific tissues or cells, including skeletal muscles. Wang et al. developed an miR-26a-overexpressing small EV that contained a muscle-specific targeting peptide, helping select delivery to muscle cells [81]. Small EVs can be injected into the muscles, ensuring localized delivery and a more specific targeting than intravenous administration [82]. Small EVs can also be delivered via ultrasound-mediated therapy. Although research on skeletal muscle has not yet been developed widely, ultrasound techniques have significantly enhanced small EV delivery to neurons through the blood–brain barrier. Using ultrasound-targeted microbubble disruption (UTMD) techniques may also improve substantially small EV delivery to muscles through enhanced endothelial cells’ permeability, increased intracellular calcium levels, and stimulated nitric oxide synthase effects, ameliorating small EV passage from vessels to organs [83,84]. These methods aim to improve small EV efficiency in targeting muscle cells. Therefore, they might efficiently improve clinical outcomes against muscle degeneration and muscle loss.

### 4.2. Mechanistic Insights: Molecular Pathways and Interaction with Muscle Cells

Guo et al. [71] conducted a study using small EVs derived from ATDSCs and found significant results regarding muscle regeneration. Their findings suggested that small EVs containing miR-125b-5p could improve muscle cell proliferation and migration and increase angiogenesis, protecting skeletal muscles against ischemic injury and decreasing muscle structural protein loss. Their intervention used the C2C12 mouse myoblast cell line and the human umbilical vein endothelial cell (HUVEC) endothelial cell line for in vitro and diabetic mice for animal experiments. The mechanisms of action involved the suppression of alkaline ceramidase 2 (ACER2) overexpression and cyclin A2, VEGF, fibronectin, and elastin upregulation. Previous studies have demonstrated that ACER2 induces autophagy and apoptosis through reactive oxygen species (ROS) generation, potentially limiting muscle regeneration [85]. Additionally, cyclins are crucial for muscle development and regeneration, improving cell cycle outcomes in various conditions [86]. Finally, it is well documented that VEGF expression upregulates muscle regeneration. Fibronectin and elastin upregulation also enhances muscle regeneration, improving skeletal muscle elasticity and endurance [87,88,89].

In a separate study using lyophilized ATDSC-derived small EVs, Mendhe et al. [72] studied the particles’ effects on a mouse model of hindlimb ischemia/reperfusion (I/R) injury. Although the results demonstrated that small EVs improved angiogenesis and reperfusion, their impact on muscle structure was adverse. Small EVs led to decreased structural protein content due to increased expression of pro-inflammatory markers in the model’s muscles. Although the intervention upregulated the expression of annexin a1, a crucial anti-inflammatory factor associated with muscle regeneration, the tumor necrosis factor alpha (TNF-α) and interleukin 6 (IL-6) pro-inflammatory cytokine levels were elevated, limiting skeletal muscle regeneration and diminishing the treatment outcomes [90].

Fu et al. [73] studied ATDSC-derived small EVs, significantly enhancing muscle bundle regeneration. By treating tendon cells and mice with cuff lesions, small EVs significantly upregulated osteogenic, chondrogenic, and tenogenesis genes, promoting the regeneration of collagen fibers and muscle bundles. Interventions are a reality regarding muscle regeneration, and most interventions modulate chondrogenic, osteogenic, and other genetic markers to modulate skeletal muscle function, even indirectly [91]. Although the authors did not directly evaluate the influence of satellite muscle cells on muscle bundle regeneration, it is worth noting that the epigenetic regulation of satellite muscle cells is crucial for muscle regeneration. In addition, epigenome alterations reduce the efficiency of muscular cells’ fate transitions required for muscle repair, contributing to muscle pathology [92]. Wang et al. [74] also studied a rat model of a massive rotator cuff tear treated with ATDSC-derived small EVs. They found that supraspinatus muscle injections significantly prevented muscular atrophy, fatty infiltration, muscle apoptosis, myotube inflammation, and enhanced vascularization. Although the mechanisms involved were discussed, the authors highlighted the decreased levels of pro-inflammatory cytokines as the mechanism of action. Regulated inflammation has a role in muscle repair and regeneration. Damaged skeletal muscle cells possess an intrinsic capacity to regenerate and repair through myogenesis by damaged myofiber-derived factors. In this context, satellite cells undergo proliferation and differentiation, fusing or combining themselves with damaged fibers to reconstitute the myotubule integrity and function. The polarization of macrophages to pro- or anti-inflammatory types are paramount in this process, in which inflammatory macrophages infiltrate damaged tissues and promote the phagocytosis of dead cells and cellular debris, facilitating satellite cell function [93]. However, uncontrolled inflammation is detrimental to muscle regeneration, as previously mentioned.

In another study, the authors utilized ATDSC-derived small EVs to treat skeletal muscle and Schwann cells and a rat model of urinary incontinence [75]. The results demonstrated enhanced muscle cell proliferation and growth, increased nerve regeneration, and muscle volume. Ni et al. demonstrated most of the molecular mechanisms from the included studies. These authors highlighted the influence of activated PI3K/Akt, Janus kinase (Jak)/signal transducer and activator of transcription (STAT), and wingless-type integration site family (Wnt) pathways, leading to the positive effects of the intervention on muscle and nerve regeneration. Previous studies have documented that PI3K/Akt signaling determines a dynamic switch between the ribonucleic acid (RNA) binding protein KSRP, leading to skeletal muscle myogenesis through opposite KSRP functions, ultimately inhibiting myogenin mRNA decay and stimulating myogenic miR maturation [94]. Targeting the Jak/STAT signaling pathway is also an interesting way to enhance muscle regeneration since this pathway is implicated in the homeostasis of several tissues. However, caution is needed, and levels of IL-6 must be strictly analyzed during the treatment due to potential deleterious effects on long-term and non-transient Jak/STAT activation [95]. Similar dual effects are attributable to the Wnt signaling pathway. Besides its importance in muscle generation during fetal myogenesis, transiently active Wnt/β-catenin signaling has not been associated with positive stem cell function during adult muscle regeneration [96].

Nguyen et al. [76] used a hindlimb ischemia mouse model to evaluate the effects of intramuscularly injected ATDSC-derived small EVs on muscle regeneration. Increased angiogenesis was associated with increased muscle structure recovery and peripheral oxygen saturation levels. At the end of the experiment, the mice recovered limb function and presented low necrosis rates. Although the authors did not evaluate the mechanisms of action involved, it is known that the small EV surface contents of CD9 and CD81 are associated with tight control of muscle cell fusion during skeletal muscle regeneration, thus regulating the fusion of myotubes, which have a vital role in the restitution of muscle architecture during muscle regeneration. In particular, CD9 depletion is associated with abnormal muscle regeneration due to uncontrolled myoblast modulation [97].

In a separate study, Jiang et al. [77] experimented with ATDSC small EVs enriched with netrin1 in vitro and in vivo using endothelial cells plus macrophages and a mouse model of ischemic limb injury, respectively. Netrin1 is a family protein with essential neuronal and osteogenic differentiation roles. In addition, this protein has been widely investigated as derived from satellite muscle cells and has a role in myogenic differentiation, especially under injury conditions [98]. It is worth noting that netrin1 knockdown in myoblasts correlates with reduced fast-type myosin heavy chain expression [98]. Jiang et al. highlighted a new signaling pathway in small EVs derived from adipose tissue therapy against muscle loss: the mitogen-activated protein kinase (MEK)/extracellular signal-regulated kinase (ERK) signaling pathway. MEK inhibited myocyte differentiation. However, ERK promoted muscle cell proliferation and migration during skeletal muscle regeneration [99,100]. Although the signaling was activated, the balance between the MEK and ERK signals was beneficial to increasing muscle cell survival and proliferation.

Finally, muscle satellite cells and mice with muscle defects were treated with ATDSC-derived small EVs in a study by Byun et al. [78]. Their results showed that the small EVs significantly induced muscle cell proliferation and myogenic differentiation and preserved the shapes and sizes of the treated muscle bundles. The molecular pathway involved was MyoD, which is essential for the activation status of muscle stem cells and promotes muscle tissue regeneration after injury [101]. Byun et al.’s results are promising as MyoD and myogenin are co-expressed during muscle regeneration. Myogenin is essential for myofiber growth and muscle stem cell homeostasis in adults [102]. However, future research is needed to evaluate the upregulated expression of MyoD in the elderly population because limited evidence is available [103].

The included studies have exhibited significant variability, which demonstrates evident limitations. The particle size and the small EV biomarkers have varied significantly among the included studies. The studied small EVs’ size range may affect the consistency and efficacy of the findings across different models and translational research regarding particle size variability. In addition, the variability in identification markers may lead to difficulties in reproduction studies across different research sites possessing different donors, which might also affect the consistency and efficacy of the results. Furthermore, the delivery methods were inconsistent, demonstrating challenges in comparing outcomes and reproducing results among studies, especially in translational research. Finally, the injection methods did not reflect real-world applications, hindering translational research and making it difficult to compare different outcomes. Additionally, while many studies have been conducted using small-scale samples, variations in experimental designs, small EV isolation methods, and small EV characterization limit the generalizability of the included results. Other essential limitations that challenge our interpretation may be the naturally occurring variance of small EVs, their proteomic and lipidomic profiles, and their interactions with the different skeletal muscles of the human body. Additionally, challenges in isolating and characterizing small EVs from different body organs may also lead to variable results and a lack of consistency in the interpretation of the findings. There is also a lack of a comprehensive analysis of potential side effects and long-term outcomes. Using animal models may be another limitation because of the differences in their physiology from humans, limiting the broad applicability and effectiveness of translational research. In terms of sarcopenia, the utilized models may not cover all aspects of sarcopenia, including its pathological nuances. Therefore, more studies on small EVs and sarcopenia-specific models must be conducted to appropriately translate findings between laboratory and clinical settings.

Finally, there are two other key difficulties in using small EVs for therapeutic interventions. Identifying and distinguishing small EVs pose significant challenges to the ongoing research in biomedical sciences due to their overlapping characteristics and complex nature in biological fluids. Their small size (<200 nm) makes it challenging to separate small EVs from other components within body fluids. Additionally, correctly identifying from which organ a separated small EV was picked also poses challenges since most of them are of endosomal origin. Proteomic and lipidomic analyses can be a pathway for identifying the origin of small EVs since organs have distinct protein and lipid interactions, which are also demonstrated within EVs. Therefore, efficient isolation and characterization are key to fully understanding small EV-based therapeutic interventions.

## 5. Clinical Implications: Potential Benefits for Sarcopenia and Muscle Loss

The evidence gathered underscores the potential of ATDSC-derived small EVs against muscular degeneration in various models and interventions. By synthesizing the existing knowledge, information regarding the use of small EVs from adipose stem cells against sarcopenia has emerged, and the effects associated with the possible beneficial effects have become evident. Small EVs from adipose stem cells promote skeletal muscle regeneration through enhanced muscle cell proliferation and migration and increased angiogenesis. Satellite cells, fibro-adipogenic progenitors, and muscle stem cells are involved in skeletal muscle development and regeneration, and the skeletal muscle’s intrinsic capacity for regeneration deteriorates with age due to cellular and extracellular changes caused by molecular deficiency and immune senescence. In sarcopenia, aged progenitor cells undergo cellular changes that lead to muscle cell proliferation and function defects [104]. In this context, augmented muscle cell proliferation and migration would be beneficial against sarcopenia since a limited and defective number of cells are available to regenerate muscle in the elderly, who are the most affected by sarcopenia. Given the possible genetic defects of aged muscle cells contributing to the development of sarcopenia and small EVs being charged with nucleic acids, addressing ATDSC-derived small EVs with miR downregulated in sarcopenia may be another avenue for future research. It is worth noting that serum miR-133a-3p and miR-200a-3p are deficient in subjects with sarcopenia, and these miRs were not evaluated in the included studies. Therefore, consistently targeting muscle cells during aging with these miRs would enhance the proliferation and differentiation of aged muscle cells [105].

Angiogenesis is essential for successful tissue repair, especially during skeletal muscle regeneration. During myogenesis, it is expected that muscle generation and angiogenesis occur together. Proper blood flow increases capillary wall tension and mechanical stress, leading to augmented sarcomere length, which is optimal for sarcopenia treatment [106]. ATDSC small EVs have also been associated with enhancing muscle perfusion. They could significantly improve oxygen flow to muscle tissues during repair, increasing oxygen and nutrient flow. This is of particular interest for sarcopenia research since not all sarcopenic patients can exercise. With aging, the capillary density of the skeletal muscles declines in a condition characterized by low expression of angiogenic factors [107]. Inflammation during aging-related muscle loss also contributes to this decrease in angiogenesis since an uncontrolled pro-inflammatory environment is deleterious to myogenic differentiation [108,109]. Therefore, the evidence suggesting the use of small EVs in sarcopenic patients is of paramount importance since enhanced angiogenesis ameliorates muscle recovery even without exercise, as previous evidence in preclinical models suggests [110].

Small EVs are also associated with skeletal muscle repair under metabolic impairment conditions, especially diabetes. The pathological connection between sarcopenia occurrence and diabetes involves poor glycemic control, oxidative stress, inflammation, and increased adiposity. This hinders muscle repair and negatively correlates with skeletal muscle mass, strength, and function during sarcopenia. In this context, diabetes can also be considered a risk factor for sarcopenia since it not only accelerates skeletal muscle mass and function loss but also correlates with sedentarism [111,112]. Small EVs are related to diabetes amelioration and increased muscle weight and fiber count. Therefore, using small EVs from stem cells may be crucial for maintaining muscle homeostasis under diabetic factors [113].

Skeletal muscle contraction is essential for the movement of the musculoskeletal system. Skeletal muscles interact with bones through tendons and ligaments to achieve movements. However, the correct position at the appropriate time during development is also essential for the body’s movement. Therefore, appropriate tendon/ligament and muscle generation and regeneration after injury are of the utmost importance. In sarcopenia, muscle function declines, and chronic tendon dysfunction increases [114]. ATDSC-derived small EVs were valuable contributors to tendon regeneration across many tendon injury models [115]. ATDSC-derived small EVs can target tendon repair. At the same time, muscle regeneration occurs since tendons and muscles function symbiotically, in a synergistic relationship [116,117]. Given that sarcopenic patients often present tendinopathies, the use of small EVs in this context is of paramount importance, as they can synergistically affect and resolve two of the most significant musculoskeletal problems of sarcopenic patients, mainly because tendinopathies lead to increased muscular dysfunction.

Skeletal muscle mass decline during aging and physical performance decline are the leading causes of muscle strength loss. Impaired motor coordination, excitation–contraction coupling, and loss of skeletal integrity are detrimental factors associated with muscle strength loss [118]. In sarcopenia, muscle mass and muscle strength are lost [119]. ATDSC-derived small EVs have also been implicated in regenerating skeletal muscle biomechanical properties, improving muscle performance, and enhancing muscle mass and strength. Additionally, small EVs have been confirmed to increase nerve regeneration, crucial for sarcopenia treatment since muscle coordination and strength are factors derived from correct innervation [120]. Therefore, using small EVs for people living with sarcopenia and neuropathies may be a crucial area for developing effective therapies for these two interrelated conditions.

## 6. Conclusions and Implications for Future Research

ATDSC-derived small EVs possess biological activities in skeletal muscle regeneration. They are a promising therapy against age-related muscle loss and sarcopenia due to relevant molecular pathway modulation as an effective intervention. However, it is worth noting that implementing ATDSC small EVs into routine, daily clinical practice remains distant, as only preclinical studies have been developed in this field. Future research must optimize delivery methods, improve mechanistic studies to overcome boundaries, and significantly translate preclinical results into randomized clinical trials. By optimizing small EV delivery methods under muscle pathological conditions, future research must guarantee the best techniques to target and modify skeletal muscle function effectively, directing therapies by using nanoparticles, small EV encapsulation, or small EV surface epitope modification. To effectively increase small EV uptake and internalization by muscles, improving small EV surface integrins, tetraspanins, and heparan sulfate proteoglycans to a more similar pattern of skeletal muscle myotubes and myoblasts may be an effective strategy [121]. Furthermore, by investigating more in depth the mechanisms by which ATDSC small EVs influence skeletal muscles’ homeostasis and, finally, improve muscle regeneration, researchers will be able to harness the full potential of small EVs by introducing specific DNA fragments, mRNAs, miR, or lncRNAs related to vital signaling pathways for muscle regeneration, based on ATDSC genetics, into small EVs as cargo. Delivering specific RNA or nucleic acids, as well as proteins and lipids related to signaling pathway activation or protein translation, will be a valuable tool for more efficient nuanced skeletal muscle regeneration, and may help overcome limitations in providing the necessary stimuli for myotubule or myoblast sequencing in generating new muscle cells. By providing muscle cells with what they need to regenerate, therapies would undoubtedly be more cost-effective, efficient, and easily translated into human research [122].

Combining therapies with small EVs plus physical exercise, nutritional supplements, or specific medications could also lead to more effective treatments against sarcopenia and age-related muscle loss. It is worth noting that dietary supplements of amino acids are a valuable strategy to counteract muscle degeneration. Medications such as sodium-glucose co-transporter 2 (SGLT2) inhibitors, growth hormone analogs, glucagon-like peptide-1 receptor agonists (GLP1-RA), metformin, statins, and dipeptidyl peptidase 4 (DPP-4) inhibitors also have a role in decreasing muscle degeneration by various mechanisms of action [123]. However, implementing combinations of therapies with small EVs must be specifically tailored and personalized to a specific population. For example, antidiabetic drugs may be poorly tolerated by non-diabetic patients due to potential adverse effects like hypoglycemia, GLP1-RA may not be appropriate for individuals suffering from inflammatory bowel diseases, and statins may not be appropriate for those with well-controlled cholesterol levels. In this context, long-term studies are necessary to monitor possible adverse effects and track if the benefits of small EVs or small EV combination therapies are sustainable in long-term interventions. Researchers must also try to elucidate the most effective doses for human-equivalent studies, overcoming limitations, such as biomarker differences among small EV interventions based on the administration route, which can be direct (e.g., intramuscular) or indirect.

Regarding personalized medicine, small EVs must be better tolerated or practical for those genetically predisposed to their action or specific cargo. In this context, genome-wide association studies (GWAS) are a qualified research method that compares genomes from many different populations to find specific genetic markers more associated with particular phenotypes, risk of disease, and response to various treatment strategies [124]. Conducting these studies on small EV therapy against muscle degeneration is of utmost importance since elderly people, who suffer more from muscle loss, may present different genetic patterns than young people, and intervening at this age is more complex [125]. Therefore, it would be more effective to deliver small EVs with a more nuanced quantity of nucleic acids or RNA-based intervention in those who are elderly, personalizing the therapeutic approach to this age group. This could also include cost-effectiveness studies since personalized treatment strategies tend to decrease costs due to increased efficacy with lower doses, reduced treatment durations, and improved outcomes.

Finally, when using small EVs in therapy, ethical and regulatory considerations must be made. These must include informed consent, since stem cells are biologically active material, and regulatory compliance, since ethical standards and legal considerations in small EV-based treatment have not yet been imposed worldwide. Equity and access considerations must also be made. Ensuring that new treatments are available and fully accessible to all patients who need them, regardless of economic or social status, is necessary to address disparities in access to advanced, improved therapies for common health concerns, like sarcopenia.

## Figures and Tables

**Figure 1 cells-14-00683-f001:**
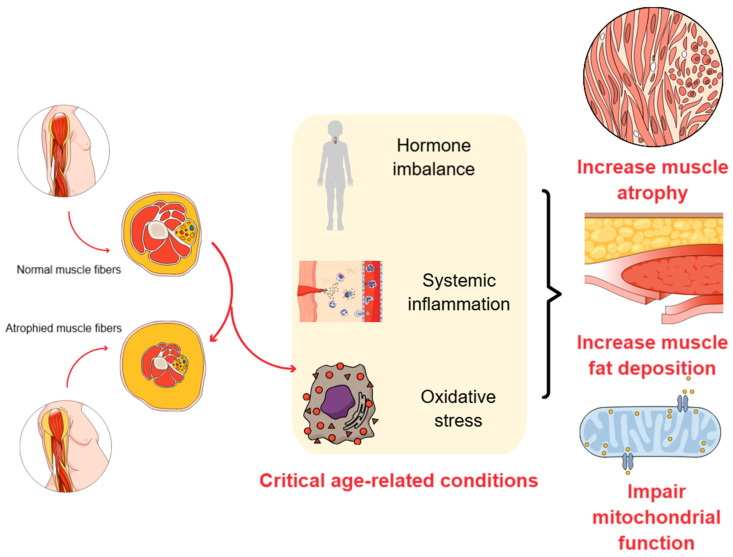
The main pathophysiological events involved in the occurrence, development, and progression of sarcopenia. Aging-related conditions, including hormone imbalance, systemic inflammation, and increased oxidative stress, lead to increased muscle atrophy, stimulated muscle fat deposition, and impaired mitochondrial function, leading to loss of muscle fibers. Created using Mind the Graph (https://mindthegraph.com/ [Accessed on 20 April 2025]).

**Figure 2 cells-14-00683-f002:**
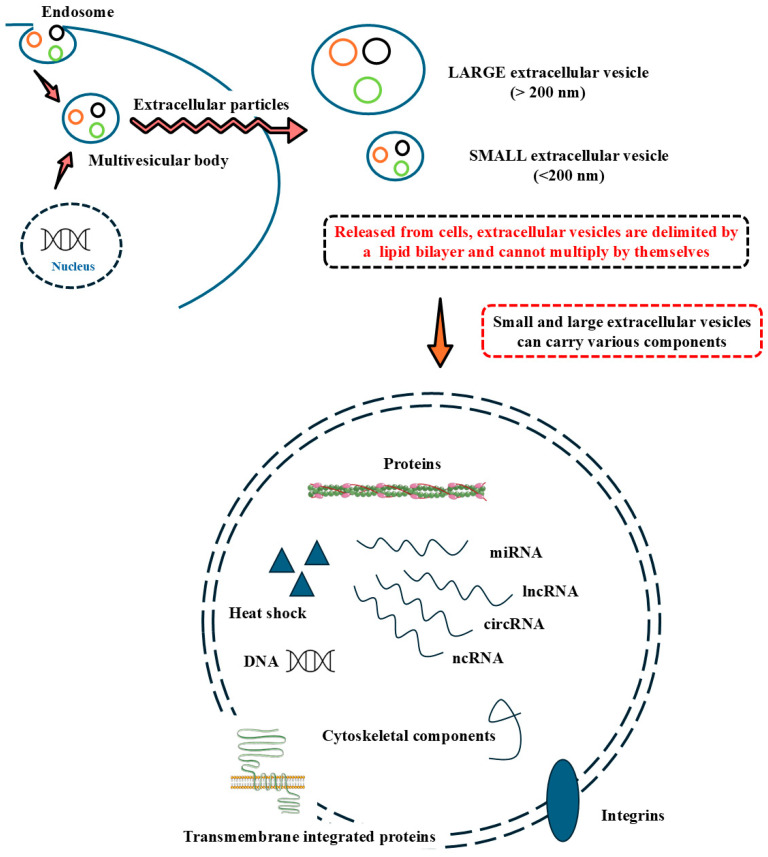
Extracellular vesicles (EVs) are cell-derived particles delimited by a lipid bilayer. They cannot multiply by themselves. They can be considered small (<200 nm) or large (>200 nm) according to their diameter. Proteins, nucleic acids, integrins, and other cellular components are loaded into EVs and released from the parent cell. Abbreviations: miRNA, micro ribonucleic acid; lncRNA, long non-coding ribonucleic acid; circRNA, circular ribonucleic acid; ncRNA, non-coding ribonucleic acid; DNA, deoxyribonucleic acid.

**Table 1 cells-14-00683-t001:** Targeting skeletal muscles with small extracellular vesicles from adipose tissue-derived stem cells: analyzing the implications for muscle regeneration.

Small EV Characteristics	Delivery Methods	Biological Functions	Mechanisms of Action	Small EV Biomarkers	Experimental Models	Ref.
miR-125 b-5p ATDSC-SEVs (159 ± 42.71 nm).	- PKH-26-labeled ATDSC-SEVs. - C2C12 co-cultured cells. - ATDSC-SEV-seeded HUVECs. - ATDSC-SEV-injected ischemic hindlimb.	- ↑Proliferation and migration of C2C12 cells and HUVECs’ angiogenesis. - ↑Muscle protection, injury repair, and vascular regeneration.	- ↓ACER2. - ↑Cyclin A2, VEGF, fibronectin, and elastin.	- CD9. - CD63. - CD81.	- C2C12 muscle and HUVEC endothelial cells. - Diabetic HLI mice.	[71]
Lyophilized ATDSC-SEVs (169 nm).	- Tail vein injection before reperfusion.	- ↑Muscle reperfusion. - However, the intervention decreased muscular structural proteins.	- ↑Annexin a1. - ↑TNF-α and IL-6.	- CD29. - CD44. - Sca-1 positive. - CD117.	- Hindlimb I/R injury mouse.	[72]
ATDSC-SEVs (~30–150 nm).	- PKH-26-labeled ATDSC-SEVs incubated with TDSCs. - Shoulder-injected ATDSC-SEV hydrogel complex.	- ↑Osteogenic and adipogenic differentiation. - ↑Rotator cuff articular segment improvement. - ↑Collagen fibers and muscle bundle regeneration.	- ↑Runx2. - ↑Sox-9. - ↑TNMD. - ↑TNC. - ↑Scx.	- CD9. - HSP70. - TSG-101.	- Human TDSCs. - Rotator cuff tear model rat.	[73]
ATDSC-SEVs (~50–150 nm).	- Supraspinatus muscle-injected small EVs.	- ↓Atrophy, fatty infiltration, apoptosis, and inflammation. - ↑Vascularization of injured muscles. - ↑Myofiber regeneration. - ↑Myofiber properties.	- ↓Inflammatory cytokines.	- CD9. - CD63. - TSG-100.	- Massive rotator cuff tear rat.	[74]
ATDSC-SEVs (~30–150 nm).	- Skeletal muscle cells and Schwann small EV-incubated cells. - Urethra-injected small EVs.	- ↑Cell growth, proliferation, and nerve regeneration. - ↑Bladder capacity and leak point pressure. - ↑Number of striated muscle fibers and peripheral nerve fibers in the urethra.	- PI3K-Akt, Jak-STAT, and Wnt modulation.	- FLOT1. - ICAM. - ALIX. - TSG-101. - ANXA5. - EpCAM. - CD63. - CD81.	- Skeletal muscle and Schwann cells. - SUI rat.	[75]
ATDSC-SEVs.	- Muscle burn site-injected small EVs.	- ↑Angiogenesis. - ↑Muscle structure recovery. - ↑Peripheral capillary oxygen. - ↑Limb function. - ↓Necrosis rate.	- Not specified.	- CD9. - CD63. - CD81.	- Hindlimb ischemia mouse.	[76]
Netrin1-enriched small EVs from ATDSC (~100–150 nm).	- CM-Dil-labeled N-Exos co-cultured with HUVEC. - N-Exos-incubated RAW264.7 cells. - Ischemic limb muscle N-Exos injection.	- ↑HUVEC proliferation, migration, and tube formation. - ↑HUVEC apoptosis resistance. - ↑Angiogenesis, collateral artery remodeling, and muscle protection. - ↓Inflammation.	- ↑PI3K/Akt/eNOS. - ↑MEK/ERK. - ↑M1 to M2 macrophage polarization.	- CD9. - CD63. - CD81. - TSG-101.	- HUVEC cells and RAW264.7 macrophages. - Diabetic limb ischemia mouse.	[77]
MSC-derived small EVs (118 nm).	- Fresh medium containing small EVs. - Muscle defects filled with small EVs and fibrin gel.	- ↑Muscle cell proliferation. - ↑Myogenic differentiation. - ↑Preservation of muscle bundles’ shapes and sizes.	- ↑Myogenin. - ↑MyoD1.	- CD9. - CD81. - CD63.	- Human skeletal muscle satellite cells. - Surgically induced muscle defect mouse.	[78]

Abbreviations: ACER2, alkaline ceramidase 2; ATDSC, adipose tissue-derived stem cell; Akt, protein kinase b; eNOS, endothelial nitric oxide synthase; ERK, extracellular signal-regulated kinase; EVs, extracellular vesicles; HLI, hindlimb ischemia; HUVEC, human umbilical vein endothelial cell; IL, interleukin; I/R, ischemia/reperfusion; Jak, Janus kinase; MEK, mitogen-activated protein kinase; miR, micro ribonucleic acid; MSCs, mesenchymal stem cells; MyoD1, myoblast determination protein 1; N-Exos, Netrin1-enriched exosomes; PI3K, phosphoinositide 3-kinase; Runx2, runt-related transcription factor 2; Scx, scleraxis; SEV, small extracellular vesicle; Sox-9, SRY-box transcription factor 9; STAT, signal transducer and activator of transcription; SUI, stress urinary incontinence; TDSCs, tendon-derived stem cells; TNC, tenascin C; TNF-α, tumor necrosis factor alpha; TNMD, tenomodulin; VEGF, vascular endothelial growth factor; Wnt, wingless-type integration site family.

## Data Availability

No new data were created or analyzed in this study. Data sharing is not applicable to this article.

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
