# Peer review of "Targeting Muscle Regeneration with Small Extracellular Vesicles from Adipose Tissue-Derived Stem Cells—A Review"

_cells, 2025, doi:10.3390/cells14100683_

Round 1

Reviewer 1 Report

Comments and Suggestions for Authors

I would have the following major points:

  • There is inconsistent use of terminology throughout the text – specifically, the terms "extracellular vesicles", “small extracellular vesicles” and "exosomes" are used interchangeably without proper clarification. The authors are encouraged to refer to the International Society for Extracellular Vesicles (ISEV) guidelines to ensure appropriate and standardized terminology. It would also be helpful to clearly define what specifically falls under the chosen term.
  • While the manuscript is centered on muscle regeneration, sarcopenia, and muscle loss, Section 2.1 addresses unrelated topics like wound healing, peritoneal fibrosis, chronic endometritis, and acne-related inflammation. As a result, this section feels off-topic and distracts from the overall focus. Additionally, the highly technical details presented from lines 133 to 149 in section 2.1 do not seem relevant and could be significantly condensed.
  • Section 3.1 presents general information on extracellular vesicles that is not specific to adipose tissue. As such this section could serve as a broader introduction to extracellular vesicles, especially if merged with section 3.2. Moreover, section 3.1 contains many technical details on extracellular vesicle purification that could be more effectively summarized for better clarity.
  • Table 1 is quite dense and could benefit from a more concise, bullet-point format to improve readability. Additionally, the studies mentioned in table 1 focus on muscle regeneration rather than addressing sarcopenia or aging. While informative, the “clinical relevance” column extends beyond the scope of the cited studies and is already discussed within the main text in section 5.
  • Multiple claims are made throughout the manuscript without proper references to support them. Non-exhaustive examples include: lines 54–57 (“Due to their ubiquity […] therapeutic functions”), lines 74–78 (“Pathologically, the size […] regenerative capacity”), lines 95-96 (“They are released […] in skeletal muscle homeostasis”), lines 220–221 (“They can regulate intercellular […] 400 clinical trials worldwide”), lines 246–248 (“However, following secretion […] until captured by target cells”), and lines 387–389 (“In addition, this protein […] especially under injury conditions”). Please ensure that all claims are supported by appropriate references to published work.

I have a few other minor remarks:

  • The title is quite long and could be more impactful if shortened for clarity and focus.
  • Figure 1 would benefit from being a general overview of the different populations of extracellular vesicles and their biogenesis. I would recommend putting it in section 3.1 for better readability, alongside the classification of extracellular vesicles.
  • Figure 2 is somewhat disconnected from the main text as it shows several signaling pathways that are not addressed in the accompanying section. An illustrative figure highlighting the interplays between sarcopenia, aging, and muscle regeneration could be of interest.
  • Incorrect terms are found in the text, including: line 66 “arranging”, line 146 “spaced ages”, line 356 “decontrolled”, line 377 “in the experiment’s final”, line 435 “side-population cells”, line 444 “fulfilling”
  • Titles of sections 3 and 4 are quite long and could be more concise and direct.
  • Extracellular vesicles should be abbreviated as EV throughout the text.

Author Response

RESPONSE TO REVIEWERS' COMMENTS

Manuscript number: cells-3575382 ― Cells (MDPI)

"Targeting Muscle Regeneration with Small Extracellular Vesicles from Adipose Tissue-derived Stem Cells – A Review "

The authors of this document wish to express their deepest gratitude to the Editor-in-Chief and the Reviewer for their thorough and insightful evaluation of our manuscript. Their expert feedback has been invaluable in enhancing the quality of our work. We have carefully considered and diligently implemented each suggestion, significantly improving the manuscript. We have made substantial revisions to address the points raised. These noteworthy changes are marked mainly with YELLOW-highlighted text throughout the document for ease of reference. A note will be provided for the referee's attention for corrections highlighted in a different color. Additionally, we have prepared a detailed and comprehensive response to each comment and suggestion. This response is organized in a "point-by-point" format below, ensuring that every concern has been thoroughly addressed and explained. We sincerely appreciate the time and effort invested by the Editor-in-Chief and the Reviewer, and we believe their contributions have significantly strengthened the final version of our manuscript.

REVIEWER #1

General response

Dear Erudite Reviewer, thank you for taking the time to revise our manuscript and for allowing us to improve based on your valuable comments and suggestions. After addressing all your comments and suggestions regarding our manuscript text, we are confident that a significantly enhanced manuscript version has emerged. We are excited to resubmit the modified version for your perusal and reevaluation. Thank you for your brilliant insights, essential contributions, and feedback. You do have an eye for improvement. As a gesture of our utmost respect for you, we would like to provide you with a detailed and comprehensive point-by-point response to your comments below. Thank you once again for your time and patience in revising our article.

Comment #1

There is inconsistent use of terminology throughout the text – specifically, the terms "extracellular vesicles", “small extracellular vesicles” and "exosomes" are used interchangeably without proper clarification. The authors are encouraged to refer to the International Society for Extracellular Vesicles (ISEV) guidelines to ensure appropriate and standardized terminology. It would also be helpful to clearly define what specifically falls under the chosen term.

Response

Dear Erudite Reviewer, thank you for this comment. You are entirely correct, and we agree that our manuscript would benefit from improvements in the abbreviation work within its text. To improve our manuscript accordingly, we followed the strict guidelines of Welsh et al. (2024) from “Minimal information for studies of extracellular vesicles (MISEV2023)” and deliberately updated some terms in our manuscript, which have been consistently highlighted using GREEN or BLUE colors within the entire manuscript. “Exosomes” have been replaced with “Small Extracellular Vesicles.” According to the guidelines abovementioned, the term “Small Extracellular Vesicles” is appropriate since these are generally Extracellular Vesicles with a diameter of less than 200 nm, which follow the scope of our included studies. To respect your suggestions, we also included Lines 50-57 on Page 2, in which we clearly define what the specific nomenclatures refer to with the utmost criteria.

            Thank you for your patience and guidance. Your recommendations have been essential for us to revise our manuscript critically.

Comment #2

While the manuscript is centered on muscle regeneration, sarcopenia, and muscle loss, Section 2.1 addresses unrelated topics like wound healing, peritoneal fibrosis, chronic endometritis, and acne-related inflammation. As a result, this section feels off-topic and distracts from the overall focus. Additionally, the highly technical details presented from lines 133 to 149 in section 2.1 do not seem relevant and could be significantly condensed.

Response

Dear Erudite Reviewer, thank you for this comment. You are entirely correct, and we agree with the points raised. Therefore, we modified the manuscript to follow your recommendations and improve our text strictly. Previous Lines 158-187 on Page 5 have been deleted. Per your brilliant suggestions, previous Lines 131-158 on Pages 4-5 have been condensed into Lines 160-178 on Pages 4-5. Section 2.1 is better organized and does not fall out of our manuscript’s scope since we do not refer to other health benefits of small extracellular vesicles than those related to muscle regeneration. Previously, Section 2.1 spanned 29 lines. Now the section spans 18 lines, which means an 11-line condensation. Since no new text was added or created for this revision, we highlighted the modifications using the PINK color.

            Again, thank you for your patience and guidance. We are thankful for the opportunity to communicate with you and for your rigorous assessment of our manuscript.

Comment #3

Section 3.1 presents general information on extracellular vesicles that is not specific to adipose tissue. As such this section could serve as a broader introduction to extracellular vesicles, especially if merged with section 3.2. Moreover, section 3.1 contains many technical details on extracellular vesicle purification that could be more effectively summarized for better clarity.

Response

Dear Erudite Reviewer, thank you for this insightful comment and brilliant input. You are entirely correct, and we agree that previous Subsection 3.1 could be condensed and merged with previous Subsection 3.2 to enrich the discussion and not impose technical details for our future readers. Therefore, we condensed previous Subsection 3.1 in Lines 206-223 on Pages 5-6, representing a huge condensation from previous Lines 212-243 on Page 6, representing a 15-line condensation. This condensed text has also been merged with previous Subsection 3.2, which now titles Section 3, which no longer contains subsections. Ultimately, we did exactly what you recommended, condensing Previous Subsection 3.1’s text and merging this previous text with previous Subsection 3.2’s text, which is now the main title of Section 3 (Lines 201-241 on Pages 5-6), named “A Mechanistic Approach to Small Extracellular Vesicle Isolation, Characterization, and Mediated Communication.” Since no new text was added or created for this revision, we highlighted the modifications using the PINK color.

            Thank you for your work assessing our manuscript and for your recommendations. We are thankful for being able to revise our manuscript based on your precious comments. Your help has been invaluable in reshaping our manuscript for the better!

Comment #4

Table 1 is quite dense and could benefit from a more concise, bullet-point format to improve readability. Additionally, the studies mentioned in table 1 focus on muscle regeneration rather than addressing sarcopenia or aging. While informative, the “clinical relevance” column extends beyond the scope of the cited studies and is already discussed within the main text in section 5.

Response

Dear Erudite Editor, thank you for bringing this to our attention. You are correct, and we agree that our Table 1 would benefit from further improvements. Therefore, we modified the columns in our table, improving content disposition using bullet points. Additionally, we deleted the previous column, “Clinical Relevance,” since we had already discussed it in the central discussion part of our manuscript. Finally, we modified Table 1’s title to refer strictly to muscle regeneration and rigorously follow the methodology of the studies we included. The new Table 1 can be found on Pages 8-9, and the new Table 1’s title can be found in Lines 257-258 on Page 8. We are confident we are presenting a significantly modified and improved version of our manuscript based on your suggestions.

            Again, thank you for everything. Your valuable comments and suggestions have improved our text’s readability and conciseness.

Comment #5

Multiple claims are made throughout the manuscript without proper references to support them. Non-exhaustive examples include: lines 54–57 (“Due to their ubiquity […] therapeutic functions”), lines 74–78 (“Pathologically, the size […] regenerative capacity”), lines 95-96 (“They are released […] in skeletal muscle homeostasis”), lines 220–221 (“They can regulate intercellular […] 400 clinical trials worldwide”), lines 246–248 (“However, following secretion […] until captured by target cells”), and lines 387–389 (“In addition, this protein […] especially under injury conditions”). Please ensure that all claims are supported by appropriate references to published work.

Response

Dear Erudite Reviewer, Thank you for this comment. You are entirely correct, and we agree with you. To improve our manuscript accordingly, we added 13 references to each fragment you provided. The first can be found in Lines 59-60 on Page 2, the second can be found in Lines 77-81 on Page 2, the third can be found in Lines 116-117 on Page 4, the fourth can be found in Lines 213-214 on Page 5, the fifth can be found in Lines 225-228 on Page 6, and the sixth can be found in Lines 375-377 on Page 11. We ensured no unreferenced information in our revised document by this comment. The complete list of references added is below. Thank you for your attention to this matter. We are thankful for the opportunity to revise our manuscript based on your comments.

  • Baumann CW, Kwak D, Liu HM, Thompson LV. Age-induced oxidative stress: how does it influence skeletal muscle quantity and quality? J Appl Physiol (1985). 2016;121(5):1047-52.
  • Bhavsar D, Raguraman R, Kim D, Ren X, Munshi A, Moore K, et al. Exosomes in diagnostic and therapeutic applications of ovarian cancer. J Ovarian Res. 2024;17(1):113.
  • Careccia G, Mangiavini L, Cirillo F. Regulation of Satellite Cells Functions during Skeletal Muscle Regeneration: A Critical Step in Physiological and Pathological Conditions. Int J Mol Sci. 2023;25(1).
  • Chen M, Wang Y, Deng S, Lian Z, Yu K. Skeletal muscle oxidative stress and inflammation in aging: Focus on antioxidant and anti-inflammatory therapy. Front Cell Dev Biol. 2022;10:964130.
  • Ferri E, Marzetti E, Calvani R, Picca A, Cesari M, Arosio B. Role of Age-Related Mitochondrial Dysfunction in Sarcopenia. Int J Mol Sci. 2020;21(15).
  • Isola AL, Chen S. Exosomes: The Messengers of Health and Disease. Curr Neuropharmacol. 2017;15(1):157-65.
  • Kosmac K, Gonzalez-Freire M, McDermott MM, White SH, Walton RG, Sufit RL, et al. Correlations of Calf Muscle Macrophage Content With Muscle Properties and Walking Performance in Peripheral Artery Disease. J Am Heart Assoc. 2020;9(10):e015929.
  • Li CJ, Fang QH, Liu ML, Lin JN. Current understanding of the role of Adipose-derived Extracellular Vesicles in Metabolic Homeostasis and Diseases: Communication from the distance between cells/tissues. Theranostics. 2020;10(16):7422-35.
  • Porcu C, Dobrowolny G, Scicchitano BM. Exploring the Role of Extracellular Vesicles in Skeletal Muscle Regeneration. Int J Mol Sci. 2024;25(11).
  • Wiedmer P, Jung T, Castro JP, Pomatto LCD, Sun PY, Davies KJA, et al. Sarcopenia - Molecular mechanisms and open questions. Ageing Res Rev. 2021;65:101200.
  • Dilsiz N. A comprehensive review on recent advances in exosome isolation and characterization: Toward clinical applications. Transl Oncol. 2024;50:102121.
  • Xu M, Feng T, Liu B, Qiu F, Xu Y, Zhao Y, et al. Engineered exosomes: desirable target-tracking characteristics for cerebrovascular and neurodegenerative disease therapies. Theranostics. 2021;11(18):8926-44.
  • Suzuki T, Mori A, Maeno T, Arimatsu R, Ichimura E, Nishi Y, et al. Abundant Synthesis of Netrin-1 in Satellite Cell-Derived Myoblasts Isolated from EDL Rather Than Soleus Muscle Regulates Fast-Type Myotube Formation. Int J Mol Sci. 2021;22(9).

Comment #6

The title is quite long and could be more impactful if shortened for clarity and focus.

Response

Dear Erudite Reviewer, thank you for this comment and suggestion. You are entirely correct, and we agree that modifying our title to be more consistent with our research focus would greatly benefit the overall readability of our findings. To improve our title and make it more impactful, we made significant modifications, recombining words and using fewer elements. Our title is “Targeting Muscle Regeneration with Small Extracellular Vesicles from Adipose Tissue-derived Stem Cells – A Review,” which is accurate and precisely reflects our research focus. The new title can be found in Lines 2-3 on Page 1 of the revised manuscript document.

            Thank you for being so collaborative with us. It is an honor to work alongside such an esteemed reviewer.

Comment #7

Figure 1 would benefit from being a general overview of the different populations of extracellular vesicles and their biogenesis. I would recommend putting it in section 3.1 for better readability, alongside the classification of extracellular vesicles.

Response

Dear Erudite Reviewer, thank you for this insightful suggestion. To improve our manuscript, we modified Figure 2, which is now presented within Section 3 of the revised manuscript, documented on Page 7, based on your guidelines and recommendations. We put information about classifying small and large extracellular vesicles in the figure. Additionally, we maintained information about their possible content. Figure 2’s legend has also been modified and updated to highlight the figure’s new content. The new legend in Lines 243-249 on Page 7 of the revised manuscript document can be found. Finally, Figure 2’s first mention has been moved to Lines 240-241 on Page 6. Thank you for this recommendation. We believe our figure has significantly improved after addressing your comments and suggestions.

Comment #8

Figure 2 is somewhat disconnected from the main text as it shows several signaling pathways that are not addressed in the accompanying section. An illustrative figure highlighting the interplays between sarcopenia, aging, and muscle regeneration could be of interest.

Response

Dear Erudite Reviewer, you are entirely correct. Thank you for bringing this to our attention. To improve our manuscript accordingly, we improved Figure 1’s content. The figure depicts the key pathophysiological pathways induced by aging in skeletal muscles, highlighting the interplay between sarcopenia, aging, and muscle regeneration and loss. The new Figure 1 can be found on Page 3. Its legend has also been updated to highlight the figure’s main content. The revised legend of Figure 1 can be found in Lines 85-89 on Page 3 of the revised manuscript document.

            Again, thank you for your guidance. Working alongside you has been a privilege, and we are eager to anticipate your positive response to the revised version of the manuscript.

Comment #9

Incorrect terms are found in the text, including: line 66 “arranging”, line 146 “spaced ages”, line 356 “decontrolled”, line 377 “in the experiment’s final”, line 435 “side-population cells”, line 444 “fulfilling”

Response

Dear Erudite Reviewer, thank you for bringing this to our attention. We agree with you that the above-mentioned are typos and should be corrected. Therefore, we implemented modifications to our manuscript’s text to ensure the utmost readability and transparency in the presentation of our findings, ensuring that the entire manuscript is accurate in its words. To demonstrate our willingness, we invite you to revise the abovementioned corrections. The first correction can be found in Line 67 on Page 2, the second correction can be found in Line 168 on Page 5, the third correction can be found in Lines 343-344 on Page 10, the fourth correction can be found in Line 3654 on Page 11, the fifth correction can be found in Line 439 on Page 12, and the sixth correction can be found in Line 448 on Page 12. Thank you for addressing these concerns for us. We are thankful for the opportunity to revise our manuscript based on your feedback.

Comment #10

Titles of sections 3 and 4 are quite long and could be more concise and direct.

Response

Dear Erudite Reviewer, thank you for this insightful suggestion. We modified Section 3’s title to “A Mechanistic Approach to Small Extracellular Vesicle Isolation, Characterization, and Mediated Communication” in Lines 201-202 on Page 5 and Section 4’s title to “Small Extracellular Vesicles from ATDSCs in Muscle Targeting” in Line 250 on Page 7. The manuscript has been significantly improved based on this suggestion. Thank you for everything!

Comment #11

Extracellular vesicles should be abbreviated as EV throughout the text.

Response

Dear Erudite Reviewer, thank you for bringing this to our attention. You are correct that using EVs, abbreviating Extracellular Vesicles, would enhance our manuscript's readability and consistency. Therefore, we modified the entire manuscript, and the abbreviations for EVs can be found in GREEN highlighted color. We EVs are accompanied by the word “small” to confer the utmost criteria for the readability of our findings. The word “small” is highlighted using a BLUE color. Thank you for your insightful comments and suggestions. Your recommendations were instrumental in reshaping our manuscript for the better.

I, the corresponding author of the manuscript "Targeting Muscle Regeneration with Small Extracellular Vesicles from Adipose Tissue-derived Stem Cells – A Review" under the assigned ID cells-3575382, on behalf of my coauthors, once again extend my heartfelt gratitude to the knowledgeable Editor-in-Chief and reviewers for their time and expertise in revising our manuscript. After we addressed their constructive and refined feedback and suggestions, a significantly improved manuscript version emerged. Undoubtedly, their insightful suggestions and feedback have significantly enhanced the quality of our manuscript. We respectfully are at the disposal of the Editor-in-Chief and the Reviewer to address any additional suggestions regarding our publication. If you are satisfied with our newly refined and significantly improved version, we look forward to the acceptance of our article for publication in this prestigious journal, Cells. Thank you once again for your time and expertise.

Reviewer 2 Report

Comments and Suggestions for Authors

The review provides a thorough outlook on the use of EVs (exosomes) for muscle regeneration and age-related muscle loss. However, it appears to omit important studies that could enhance its comprehensiveness. For instance, Sanz-Ros et al., 2022 provide insights into EV-associated regeneration of muscle tissues and prevents frailty. Another study (Zhou et al., 2025) shows the mitigation of chronic inflammation that plays important role in aging. Including these references would strengthen this manuscript.

References:

  1. Sanz-Ros J, Romero-García N, Mas-Bargues C, et al. Small extracellular vesicles from young adipose-derived stem cells prevent frailty, improve health span, and decrease epigenetic age in old mice. Sci Adv. 2022;8(42):eabq2226. doi:10.1126/sciadv.abq2226
  2. Zhou Q, Gao J, Wu G, et al. Adipose progenitor cell-derived extracellular vesicles suppress macrophage M1 program to alleviate midlife obesity. Nat Commun. 2025;16(1):2743. Published 2025 Mar 20. doi:10.1038/s41467-025-57444-y

Author Response

RESPONSE TO REVIEWERS' COMMENTS

Manuscript number: cells-3575382 ― Cells (MDPI)

"Targeting Muscle Regeneration with Small Extracellular Vesicles from Adipose Tissue-derived Stem Cells – A Review "

The authors of this document wish to express their deepest gratitude to the Editor-in-Chief and the Reviewer for their thorough and insightful evaluation of our manuscript. Their expert feedback has been invaluable in enhancing the quality of our work. We have carefully considered and diligently implemented each suggestion, significantly improving the manuscript. We have made substantial revisions to address the points raised. These noteworthy changes are marked mainly with YELLOW-highlighted text throughout the document for ease of reference. A note will be provided for the referee's attention for corrections highlighted in a different color. Additionally, we have prepared a detailed and comprehensive response to each comment and suggestion. This response is organized in a "point-by-point" format below, ensuring that every concern has been thoroughly addressed and explained. We sincerely appreciate the time and effort invested by the Editor-in-Chief and the Reviewer, and we believe their contributions have significantly strengthened the final version of our manuscript.

REVIEWER #2

General Comment

The review provides a thorough outlook on the use of EVs (exosomes) for muscle regeneration and age-related muscle loss.

General response

Dear Erudite Reviewer, thank you for taking the time to revise our manuscript and for allowing us to improve based on your valuable comments and suggestions. After addressing all your comments and suggestions regarding our manuscript text, we are confident that a significantly enhanced manuscript version has emerged. We are excited to resubmit the modified version for your perusal and reevaluation. Thank you for your brilliant insights, essential contributions, and feedback. You do have an eye for improvement. As a gesture of our utmost respect for you, we would like to provide you with a detailed and comprehensive point-by-point response to your comments below. Thank you once again for your time and patience in revising our article.

Comment #1

However, it appears to omit important studies that could enhance its comprehensiveness. For instance, Sanz-Ros et al., 2022 provide insights into EV-associated regeneration of muscle tissues and prevents frailty. Another study (Zhou et al., 2025) shows the mitigation of chronic inflammation that plays important role in aging. Including these references would strengthen this manuscript.

References:

Sanz-Ros J, Romero-García N, Mas-Bargues C, et al. Small extracellular vesicles from young adipose-derived stem cells prevent frailty, improve health span, and decrease epigenetic age in old mice. Sci Adv. 2022;8(42):eabq2226. doi:10.1126/sciadv.abq2226

Zhou Q, Gao J, Wu G, et al. Adipose progenitor cell-derived extracellular vesicles suppress macrophage M1 program to alleviate midlife obesity. Nat Commun. 2025;16(1):2743. Published 2025 Mar 20. doi:10.1038/s41467-025-57444-y

Response

Dear Erudite Reviewer, thank you for this insightful comment. We agree that adding key references to our manuscript would enhance its readability and quality. To improve our manuscript accordingly, we cited the article by Zhou et al. in Lines 100-106 on Page 3 of the revised manuscript document alongside some newly added text. EVs from white adipose tissue progenitor cells show impaired ability to reduce inflammaging in adipose tissue macrophages. In midlife patients, adipose tissue EVs lacking miR-145-5p do not suppress L-selectin in macrophages, enhancing their M1 polarization via the NF-κB pathway. In contrast, EVs from young adipose tissue stem cells effectively inhibit M1 macrophage polarization, positively influencing aging. This information has been described in our main text in the abovementioned lines and pages.

            Additionally, we cited the article by Sanz-Ros et al. in Lines 118-122 on Page 4 of our revised manuscript document. These authors demonstrated that small EVs from young adipose tissue stem cells prevent frailty, lower epigenetic age, and enhance health span in old mice. These EVs induced pro-regenerative effects and reduced oxidation, inflammation, and senescence markers in the muscles of older mice. This information has been described in our main text in the abovementioned lines and pages.

            Again, thank you for your kind and critical evaluation of our work. Your recommendations have been invaluable to enhancing our manuscript’s quality and readability. Thank you for everything!

I, the corresponding author of the manuscript "Targeting Muscle Regeneration with Small Extracellular Vesicles from Adipose Tissue-derived Stem Cells – A Review" under the assigned ID cells-3575382, on behalf of my coauthors, once again extend my heartfelt gratitude to the knowledgeable Editor-in-Chief and reviewers for their time and expertise in revising our manuscript. After we addressed their constructive and refined feedback and suggestions, a significantly improved manuscript version emerged. Undoubtedly, their insightful suggestions and feedback have significantly enhanced the quality of our manuscript. We respectfully are at the disposal of the Editor-in-Chief and the Reviewer to address any additional suggestions regarding our publication. If you are satisfied with our newly refined and significantly improved version, we look forward to the acceptance of our article for publication in this prestigious journal, Cells. Thank you once again for your time and expertise.

Reviewer 3 Report

Comments and Suggestions for Authors

Major comment:

The field of exosomes has been very popular and still growing mysteriously. A number of articles, mainly reviews are pressing theories instead of evaluating evidence and there is a growing tendency to bring chaos into clarity instead of introducing clarity into chaos. This article although tries to give a summary of a special fragment of the field does not make bolt attempts to step forward from the crowd. I have find no reference how to the nomenclature and identification is conform with the followings of MISEV (see latest ref. PMID: 39625409). Auto association and self-formation is not discussed among the possibilities of “exosome” developing. Also clear evidence for observed (not proposed) secretion of exosomes by ATDSCs is not detailed and critically interpreted. If the authors could devote more to this aspect, it had improved their review significantly.

Other remarks:

49-58: ref. 1-3 contradiction in the literature about using extracell. vesicl. or exosome expression; ref. 4. is not entirely cited relevant, in support of this no exosomes or extracellular vesicles are mentioned in its text

Fig. 1: More detailed explanation is needed in legend, , i.e. ncRNA

68: individual quality of life quality (?) - please eliminate redundancy

72: have multi-morbidity and/or frailty – could you detail more what it means?

Fig. 2: Explain the acronyms of signal molecules/pathways

100: “adipose stem cell derived exosomes”??? in a “comprehensive” review? why not EVs? see PMID: 35806052

107 “systematically evaluate the effect of exosomes” why not the verification of exosome quality?

112-127: is the source quality of ATDSC matters? Such a difference has been reported from skeletal muscles, see ref 25

161-163: ref 28 cited in a wrong way exaggerating exosomes present (secretomes or exosomes) the same occurs later in the text of that chapter

240: what is CD, Rea?

407: The paragraph started here details the problems of variations in conditions applied but does not detail the problems of identification of exosomes and distinction from exosomes (MISEV)

418-426: elaborating on lack of consistency – isn’t it exist because of natural variance, problem of reproducible isolation of “exosomes” and variation of the hundreds of muscles?

481-87: tendons and muscle are indeed coupled and even so much that improving muscle regeneration can help tendon repair and vice versa since the stretch is a stimulating factor for both even if the “exosomes” are not effecting them at all

488-89: this sentence is pathetic since muscle loss and weakening is known not just by recently growing evidence but has been recognized since centuries

500-501: “It became evident…” after having read the article I did not managed to recall any evidence (only surpassing citations) that the ATDSC derived exosomes are not EVs or others, please see major remarks

Author Response

RESPONSE TO REVIEWERS' COMMENTS

Manuscript number: cells-3575382 ― Cells (MDPI)

"Targeting Muscle Regeneration with Small Extracellular Vesicles from Adipose Tissue-derived Stem Cells – A Review "

The authors of this document wish to express their deepest gratitude to the Editor-in-Chief and the Reviewer for their thorough and insightful evaluation of our manuscript. Their expert feedback has been invaluable in enhancing the quality of our work. We have carefully considered and diligently implemented each suggestion, significantly improving the manuscript. We have made substantial revisions to address the points raised. These noteworthy changes are marked mainly with YELLOW-highlighted text throughout the document for ease of reference. A note will be provided for the referee's attention for corrections highlighted in a different color. Additionally, we have prepared a detailed and comprehensive response to each comment and suggestion. This response is organized in a "point-by-point" format below, ensuring that every concern has been thoroughly addressed and explained. We sincerely appreciate the time and effort invested by the Editor-in-Chief and the Reviewer, and we believe their contributions have significantly strengthened the final version of our manuscript.

REVIEWER #3

General Comment

The field of exosomes has been very popular and still growing mysteriously. A number of articles, mainly reviews are pressing theories instead of evaluating evidence and there is a growing tendency to bring chaos into clarity instead of introducing clarity into chaos. This article although tries to give a summary of a special fragment of the field does not make bolt attempts to step forward from the crowd.

General response

Dear Erudite Reviewer, thank you for taking the time to revise our manuscript and for allowing us to improve based on your valuable comments and suggestions. After addressing all your comments and suggestions regarding our manuscript text, we are confident that a significantly enhanced manuscript version has emerged. We are excited to resubmit the modified version for your perusal and reevaluation. Thank you for your brilliant insights, essential contributions, and feedback. You do have an eye for improvement. As a gesture of our utmost respect for you, we would like to provide you with a detailed and comprehensive point-by-point response to your comments below. Thank you once again for your time and patience in revising our article.

Comment #1

I have find no reference how to the nomenclature and identification is conform with the followings of MISEV (see latest ref. PMID: 39625409). Auto association and self-formation is not discussed among the possibilities of “exosome” developing. Also clear evidence for observed (not proposed) secretion of exosomes by ATDSCs is not detailed and critically interpreted. If the authors could devote more to this aspect, it had improved their review significantly.

Response

Dear Erudite Reviewer, thank you for this insightful suggestion. You are entirely correct in the points you raised, and we agree with you that a nuanced discussion of the parameters mentioned above would undoubtedly enhance our manuscript quality and readability. Therefore, we discussed in Lines 50-57 on Page 2 the nomenclature following Miceli et al. and MISEV 2023 for EVs, which are divided into small (< 200 nm) and large (> 200 nm). Miceli et al. have been cited within the lines above. Additionally, the evidence for small EVs isolation from adipose tissue-derived stem cells has been mentioned and interpreted in Lines 107-114 on Pages 3-4 of the revised manuscript document. Stem cells, particularly mesenchymal ones from adipose tissue, are a key source of small EVs. Due to their abundance, adipose tissue stem cells are crucial for isolating these small EVs, which play an essential role in paracrine signaling and cell communication. The process involves isolating and culturing the stem cells, followed by centrifugation to separate the small EVs from cellular debris, ensuring the retention of the small EVs secreted by these cells.

            Again, thank you for your critical evaluation of our work. Your recommendations have been invaluable in improving the quality of our manuscript. Thank you for everything!

Comment #2

49-58: ref. 1-3 contradiction in the literature about using extracell. vesicl. or exosome expression; ref. 4. is not entirely cited relevant, in support of this no exosomes or extracellular vesicles are mentioned in its text.

Response

Dear Erudite Reviewer, thank you for bringing this to our attention. To improve our manuscript, we diligently excluded previous reference n° 4 from our reference list. Additionally, although earlier references 1-3 have been retained, we diligently included the information that the EV classifications are from the Minimal Information for Studies of Extracellular Vesicles (MISEV) guidelines. You can find the updated paragraph in Lines 49-62 on Page 2. Additionally, we added four new references to our section, all citing EVs and MISEV. These are listed below for your kind review. Thank you for your cooperation and guidance. Your directions reshaped our text for the better.

Newly added references

Samuels, M.; Giamas, G. MISEV2023: Shaping the Future of EV Research by Enhancing Rigour, Reproducibility and Transparency. Cancer Gene Ther 2024, 31, 649-651, doi:10.1038/s41417-024-00759-7.

Upadhya, D.; Shetty, A.K. MISEV2023 provides an updated and key reference for researchers studying the basic biology and applications of extracellular vesicles. Stem Cells Transl Med 2024, 13, 848-850, doi:10.1093/stcltm/szae052.

Xiong, M.; Zhang, Q.; Hu, W.; Zhao, C.; Lv, W.; Yi, Y.; Wu, Y.; Wu, M. Exosomes From Adipose-Derived Stem Cells: The Emerging Roles and Applications in Tissue Regeneration of Plastic and Cosmetic Surgery. Front Cell Dev Biol 2020, 8, 574223, doi:10.3389/fcell.2020.574223.

Zhang, Y.; Lan, M.; Chen, Y. Minimal Information for Studies of Extracellular Vesicles (MISEV): Ten-Year Evolution (2014–2023). Pharmaceutics 2024, 16, 1394.

Comment #3

Fig. 1: More detailed explanation is needed in legend, , i.e. ncRNA.

Response

Dear Erudite Reviewer, thank you for this comment. Unfortunately, Reviewer #1 did not approve the previous Figure 1 for publication. Therefore, to respond to Reviewer #1, we substituted the previous Figure 1 for a new and revised Figure 2 on Page 7 of the revised manuscript document. This figure delves more into the intricate classifications of MISEV within EVs and provides more information in these graphs. However, we provided more detailed explanations in our legend in Lines 243-249 on Page 7 to improve our manuscript and respond to you, Esteemed Reviewer. Our manuscript has significantly improved after addressing your comments and recommendations. Thank you for your patience and guidance.

Comment #4

68: individual quality of life quality (?) - please eliminate redundancy.

Response

Dear Erudite Reviewer, thank you for this comment. We eliminated the redundancy. The new and revised phrase can be found in Lines 68-69 on Page 2. Thank you for your patience and guidance.

Comment #5

72: have multi-morbidity and/or frailty – could you detail more what it means?.

Response

Dear Erudite Reviewer, thank you for this comment. We delved more into the ideas we expected to pass. The new and revised phrase can be found in Lines 73-75 on Page 2. Thank you for your patience and guidance.

Comment #6

Fig. 2: Explain the acronyms of signal molecules/pathways.

Response

Dear Erudite Reviewer, thank you for this comment. Unfortunately, Reviewer #1 did not approve the previous Figure 2 for publication. Therefore, to respond to Reviewer #1, we substituted the previous Figure 1 for a new and revised Figure 1 on Page 3 of the revised manuscript document. This figure provides an overview of the pathophysiological events in sarcopenia. Aging-related factors like hormone imbalance, systemic inflammation, and oxidative stress contribute to muscle atrophy, fat deposition, and impaired mitochondrial function, resulting in muscle fiber loss. The figure’s legend can be found in Lines 85-89 on Page 3 of the revised manuscript document.

            We hope you can approve this figure. Thank you for your patience and guidance throughout this peer-review process. We are confident that your recommendations significantly contributed to the overall quality of our text.

Comment #7

100: “adipose stem cell derived exosomes”??? in a “comprehensive” review? why not EVs? see PMID: 35806052.

Response

Dear Erudite Reviewer, thank you for this comment. We improved the sentence by including EVs. The new and revised phrase can be found in Lines 124-126 on Page 4. To improve our manuscript accordingly, we followed the strict guidelines of MISEV and deliberately updated some terms in our manuscript, which have been consistently highlighted using GREEN or BLUE colors within the entire manuscript. “Exosomes” have been replaced with “Small Extracellular Vesicles.” “Extracellular Vesicles” have been abbreviated to “EVs.” “Small Extracellular Vesicles.” have been abbreviated to  “Small EVs.” Thank you for your patience and guidance. Your recommendations significantly improved our manuscript quality.

Comment #8

107 “systematically evaluate the effect of exosomes” why not the verification of exosome quality?

Response

Dear Erudite Reviewer, thank you for this comment. We modified the sentence, which is now presented in Lines 132-133 on Page 4, to ensure the utmost readability and writing quality. Thank you for everything!

Comment #9

112-127: is the source quality of ATDSC matters? Such a difference has been reported from skeletal muscles, see ref 25.

Response

Dear Erudite Reviewer, thank you for this insightful comment. We understand that we must delve into this intricate mechanism in our revised manuscript, since the quality of stem cells from the adipose tissue matters. Please find updated information in Lines 152-158 on Page 4 of our revised document. The quality of the donor is crucial for the success of anti-aging cellular therapies. Wang et al. found that young-donor adipose tissue stem cells enhanced anti-aging effects in aged mice by reducing immune cells and inflammation, a benefit not seen with old donors. Similarly, Rodriguez et al. demonstrated that young-donor fast-adherent adipose stem cells had greater regenerative effects on muscles compared to old-donor slow-adherent stem cells. Thank you for your commitment to ensuring a high-quality peer review for our article. Thank you for everything!

Comment #10

161-163: ref 28 cited in a wrong way exaggerating exosomes present (secretomes or exosomes) the same occurs later in the text of that chapter.

Response

Dear Erudite Reviewer, thank you for this comment. Due to constraints with Reviewer #1's evaluation of our manuscript, we are unable to maintain Subsection 2.1 in its entirety. Therefore, we removed much content from this subsection. You can find the updated subsection in Lines 160-178 on Pages 4-5. We deleted many exaggerated technical details for the exosome mechanistic insights into regenerative medicine, which resulted in the exclusion of some references, too, including previous reference n° 28 from Al-Ghadban (2021) that you cited above. We appreciate your understanding. We hope you can approve the new and revised Subsection 2.1 of the revised manuscript document. Thank you for everything!

Comment #11

240: what is CD, Rea?.

Response

Dear Erudite Reviewer, thank you for this comment. We updated the abbreviations in Lines 221-222 on Page 6. Cluster of differentiation refers to “CD”, and recombinant engineered antibody control refers to “Rea”. Thank you for your patience and guidance and for bringing this to our attention. You have an incredible eye for detail!

Comment #12

407: The paragraph started here details the problems of variations in conditions applied but does not detail the problems of identification of exosomes and distinction from exosomes (MISEV).

Response

Dear Erutide Reviewer, thank you for bringing this to our attention. We added Lines 420-430 on Page 12 to elucidate the problematizations you raised above fully. Thank you for your attention to detail and eye for improvement. In our newly added text, we tried to exemplify whether the identification and correct isolation of small EVs from body fluids may pose significant challenges. We also mentioned proteomic and/or lipidomic as potential tools for elucidating these concerns.

Comment #13

418-426: elaborating on lack of consistency – isn’t it exist because of natural variance, problem of reproducible isolation of “exosomes” and variation of the hundreds of muscles?

Response

Dear Erudite Reviewer, thank you for this insightful suggestion. To improve our manuscript accordingly, we elaborated on your ideas within Lines 408-413 on Page 12 of the revised manuscript document. We hope you can approve our modifications. Thank you for being so cooperative with us! It is an honor to work alongside you. Thank you for everything!

Comment #14

481-87: tendons and muscle are indeed coupled and even so much that improving muscle regeneration can help tendon repair and vice versa since the stretch is a stimulating factor for both even if the “exosomes” are not effecting them at all.

Response

Dear Erudite Editor, thank you for this important comment. To improve our manuscript accordingly, we added Lines 486-488 on Page 13 to our paragraph. Thank you for being so collaborative with us. We hope you can approve our manuscript now! We added to additional references to reference our point.

Comment #15

488-89: this sentence is pathetic since muscle loss and weakening is known not just by recently growing evidence but has been recognized since centuries.

Response

Dear Erudite Reviewer, thank you for bringing this to our attention. The sentence has been updated in Lines 493-494 on Page 13.

Comment #16

500-501: “It became evident…” after having read the article I did not managed to recall any evidence (only surpassing citations) that the ATDSC derived exosomes are not EVs or others, please see major remarks.

Response

Dear Erudite Reviewer, Thank you for bringing this to our attention. The sentence has been updated in Lines 505-506 on Page 13. Now, we start the sentence with “Small EVs possess biological activities in skeletal muscle regeneration.” Respecting MISEV is an essential aspect of our work. By replacing the word “Exosomes” with “Small EVs,” we not only respect MISEV but also depict important rationale behind the conductance of our review. All the included studies reported EVs’ diameters below 200 nm (< 200 nm). Additionally, their methodologies illustrate addressing small EVs derived from adipose tissue stem cells. Therefore, we believe our sentence is now meaningful after the modifications we have just made. Thank you for your attention to detail and eye for improvement! We are confident that our manuscript has significantly improved after addressing your corrections and recommendations. We are thankful for the opportunity to work alongside you!

I, the corresponding author of the manuscript "Targeting Muscle Regeneration with Small Extracellular Vesicles from Adipose Tissue-derived Stem Cells – A Review" under the assigned ID cells-3575382, on behalf of my coauthors, once again extend my heartfelt gratitude to the knowledgeable Editor-in-Chief and reviewers for their time and expertise in revising our manuscript. After we addressed their constructive and refined feedback and suggestions, a significantly improved manuscript version emerged. Undoubtedly, their insightful suggestions and feedback have significantly enhanced the quality of our manuscript. We respectfully are at the disposal of the Editor-in-Chief and the Reviewer to address any additional suggestions regarding our publication. If you are satisfied with our newly refined and significantly improved version, we look forward to the acceptance of our article for publication in this prestigious journal, Cells. Thank you once again for your time and expertise.

Round 2

Reviewer 1 Report

Comments and Suggestions for Authors
  • All major points have been addressed and revised accordingly.

I still have a few minor points:

  • Appropriate references to published work are missing in lines 92-96 (“However, in individuals suffering from […] adipose tissue dysfunction and sarcopenia”) and lines 426-428 (“Proteomic and lipidomic analyses […] which are also demonstrated within EVs”).
  • Several terms throughout the manuscript are used inaccurately, including: line 52 (“ultimately”), line 53 (“However”), line 92 (“However”), line 107 (“paramount”), line 109 (“communicated”), line 113 (“sustaining the observed, not proposed, small EV secretion”), line 139 (“when received […] may”), lines 145-146 (“which predisposes to the most promising populations”), line 163 (“to the harvest of ATDSCs”), line 183 (“increasing”), line 191 (“MyoD”), line 207 (“endowed”), line 244 (“However”), lines 297-298 (“small EVs derived from ATDSCs-derived small EVs”), line 324 (“phenotypic intervention”), line 328 (“whether”), line 413 (“samples”), line 420 (“Worth”), line 459 (“in-repair”), line 479 (“algorithm”), lines 120/458/469/499 (“The small EVs”).
  • Please correct: line 199 (“the effectiveness of a stem plus mature cells combination” instead of combination therapy), lines 269-270 (“highlighting small EV superiority over conventional drug delivery systems”: this has not been demonstrated), line 431 (“the most evident”)
  • Please rephrase lines 515-518 for better clarity (“Improving small EV-surface integrins […] strategy for small EV uptake and internalization by muscles”).

Author Response

RESPONSE TO REVIEWERS' COMMENTS

Manuscript number: cells-3575382 ― Cells (MDPI)

"Targeting Muscle Regeneration with Small Extracellular Vesicles from Adipose Tissue-derived Stem Cells – A Review "

The authors of this document wish to express their deepest gratitude to the Editor-in-Chief and the Reviewer for their thorough and insightful evaluation of our manuscript. Their expert feedback has been invaluable in enhancing the quality of our work. We have carefully considered and diligently implemented each suggestion, significantly improving the manuscript. We have made substantial revisions to address the points raised. These noteworthy changes are marked mainly with YELLOW-highlighted text throughout the document for ease of reference. A note will be provided for the referee's attention for corrections highlighted in a different color. Additionally, we have prepared a detailed and comprehensive response to each comment and suggestion. This response is organized in a "point-by-point" format below, ensuring that every concern has been thoroughly addressed and explained. We sincerely appreciate the time and effort invested by the Editor-in-Chief and the Reviewer, and we believe their contributions have significantly strengthened the final version of our manuscript.

REVIEWER #1

General comment

All major points have been addressed and revised accordingly.

I still have a few minor points.

General response

Dear Erudite Reviewer, thank you for your time in revising our manuscript and for your valuable feedback. We have addressed all your comments and believe the revised version is significantly improved. We appreciate your insightful contributions and are excited to resubmit the modified manuscript for your review. Below is our detailed point-by-point response to your comments. Thank you again for your support.

Comment #1

Appropriate references to published work are missing in lines 92-96 (“However, in individuals suffering from […] adipose tissue dysfunction and sarcopenia”) and lines 426-428 (“Proteomic and lipidomic analyses […] which are also demonstrated within EVs”).

Response

Dear Erudite Reviewer, thank you for bringing this to our attention. We included five additional references in Lines 92-96 on Page 3 of the revised manuscript document. The included references are below for your review. Thank you for your valuable feedback, which significantly improved our text!

References included:

  • Johnston, E.K.; Abbott, R.D. Adipose Tissue Paracrine-, Autocrine-, and Matrix-Dependent Signaling during the Development and Progression of Obesity. Cells 2023, 12, doi:10.3390/cells12030407.
  • Richard, A.J.; White, U.; Elks, C.M.; Stephens, J.M. Adipose Tissue: Physiology to Metabolic Dysfunction. In Endotext, Feingold, K.R., Ahmed, S.F., Anawalt, B., Blackman, M.R., Boyce, A., Chrousos, G., Corpas, E., de Herder, W.W., Dhatariya, K., Dungan, K., et al., Eds.; MDText.com, Inc.
  • Rohm, T.V.; Meier, D.T.; Olefsky, J.M.; Donath, M.Y. Inflammation in obesity, diabetes, and related disorders. Immunity 2022, 55, 31–55, doi:10.1016/j.immuni.2021.12.013.
  • Sheptulina, A.F.; Antyukh, K.Y.; Kiselev, A.R.; Mitkovskaya, N.P.; Drapkina, O.M. Possible Mechanisms Linking Obesity, Steroidogenesis, and Skeletal Muscle Dysfunction. Life (Basel) 2023, 13, doi:10.3390/life13061415.
  • Wang, T. Searching for the link between inflammaging and sarcopenia. Ageing Res Rev 2022, 77, 101611, doi:10.1016/j.arr.2022.101611.

Comment #2

Several terms throughout the manuscript are used inaccurately, including: line 52 (“ultimately”), line 53 (“However”), line 92 (“However”), line 107 (“paramount”), line 109 (“communicated”), line 113 (“sustaining the observed, not proposed, small EV secretion”), line 139 (“when received […] may”), lines 145-146 (“which predisposes to the most promising populations”), line 163 (“to the harvest of ATDSCs”), line 183 (“increasing”), line 191 (“MyoD”), line 207 (“endowed”), line 244 (“However”), lines 297-298 (“small EVs derived from ATDSCs-derived small EVs”), line 324 (“phenotypic intervention”), line 328 (“whether”), line 413 (“samples”), line 420 (“Worth”), line 459 (“in-repair”), line 479 (“algorithm”), lines 120/458/469/499 (“The small EVs”).

Response

Dear Erudite Reviewer, thank you for these comments and recommendations. We modified all the sentences you mentioned to improve our manuscript’s quality and readability. The changes are highlighted in Lines 52-53 on Page 2, Lines 53-54 on Page 2, Lines 92-96 on Page 3, Line 107 on Page 3, Lines 108-111 on Page 3, Lines 111-113 on Pages 3-4, Lines 119-120 on Page 4, Lines 137-140 on Page 4, Lines 142-144 on Page 4, Lines 161-165 on Pages 4-5, Lines 180-181 on Page 5, Lines 188-190 on Page 5, Lines 205-206 on Page 5, Lines 241-242 on Page 7, Lines 291-292 on Page 9, Lines 292-295 on 406-407 on Page 11, Lines 413-414 on Page 12, Lines 451-452 on Page 12, Lines 461-462 on Page 13, Lines 471-472 on Page 13, and Lines 490-492 on Page 13. Thank you for being so cooperative! Your comments have been invaluable in improving our text. We are grateful for the opportunity to communicate with such a critical reviewer.

Comment #3

Please correct: line 199 (“the effectiveness of a stem plus mature cells combination” instead of combination therapy), lines 269-270 (“highlighting small EV superiority over conventional drug delivery systems”: this has not been demonstrated), line 431 (“the most evident”).

Response

Dear Erudite Reviewer, thank you for this insightful suggestion. We appreciate your commitment to ensuring a critical peer-review process and are thankful for the opportunity to revise our manuscript based on your comments. To improve our manuscript, we diligently modified the lines you raised above. The modifications are highlighted in Lines 194-198 on Page 5, Line 265 on Page 9, and Line 424 on Page 12.

Comment #4

Please rephrase lines 515-518 for better clarity (“Improving small EV-surface integrins […] strategy for small EV uptake and internalization by muscles”).

Response

Dear Erudite Reviewer, thank you for being so cooperative and improving our manuscript accordingly. We agree with you that the abovementioned phrase would benefit from further editing. Therefore, we implemented modifications in Lines 506-509 on Page 13. Thank you for your attention to this matter!

I, the corresponding author of the manuscript "Targeting Muscle Regeneration with Small Extracellular Vesicles from Adipose Tissue-derived Stem Cells – A Review" (ID cells-3575382), would like to express my gratitude to the Editor-in-Chief and reviewers for their valuable feedback. We have carefully addressed their suggestions, leading to a significantly improved manuscript. We are open to any further input from the Editor-in-Chief and reviewers. If you are satisfied with our revised version, we look forward to the acceptance of our article in Cells. Thank you for your time and expertise.

Reviewer 3 Report

Comments and Suggestions for Authors

The authors have amended their manuscript significantly and answered all the comments and requests. I think the paper is suitable for publication.

Author Response

Dear Erudite Reviewer, thank you for revising our manuscript and for your valuable feedback. Your comments significantly improved our text. We appreciate your insightful contributions and are excited to publish the modified manuscript. Thank you for being so supportive.